# Conflict-Aware Knowledge Editing in the Wild: Semantic-Augmented Graph Representation for Unstructured Text

**Zhange Zhang**[1,2,3]*  **Zhicheng Geng**[1,2]*  **Yuqing Ma**[1,2]†
**Tianbo Wang**[2,4]  **Kai Lv**[5]  **Xianglong Liu**[2,4]

Institute of Artificial Intelligence, Beihang University[1]
State Key Laboratory of Complex & Critical Software Environment, Beihang University[2]
Beijing Advanced Innovation Center for Future Blockchain and Privacy Computing[3]
School of Computer Science and Engineering, Beihang University[4], Beijing Jiaotong University[5]
{zhangezhang,mayuqing}@buaa.edu.cn

## Abstract

Large Language Models (LLMs) have demonstrated broad applications but suffer from issues like hallucinations, erroneous outputs and outdated knowledge. Model editing emerges as an effective solution to refine knowledge in LLMs, yet existing methods typically depend on structured knowledge representations. However, real-world knowledge is primarily embedded within complex, unstructured text. Existing structured knowledge editing approaches face significant challenges when handling the entangled and intricate knowledge present in unstructured text, resulting in issues such as representation ambiguity and editing conflicts. To address these challenges, we propose a Conflict-Aware Knowledge Editing in the Wild (CAKE) framework, the first framework explicitly designed for editing knowledge extracted from wild unstructured text. CAKE comprises two core components: a Semantic-augmented Graph Representation module and a Conflict-aware Knowledge Editing strategy. The Semantic-augmented Graph Representation module enhances knowledge encoding through structural disambiguation, relational enrichment, and semantic diversification. Meanwhile, the Conflict-aware Knowledge Editing strategy utilizes a graph-theoretic coloring algorithm to disentangle conflicted edits by allocating them to orthogonal parameter subspaces, thereby effectively mitigating editing conflicts. Experimental results on the AKEW benchmark demonstrate that CAKE significantly outperforms existing methods, achieving a 15.43% improvement in accuracy on llama3 editing tasks. Our framework successfully bridges the gap between unstructured textual knowledge and reliable model editing, enabling more robust and scalable updates for practical LLM applications.

## 1 Introduction

Large Language Models (LLMs) exhibit remarkable knowledge retention capabilities and have been widely deployed in applications such as conversational agents [1, 2], medical diagnosis [3], and code generation [4]. Nevertheless, several fundamental challenges persist in current LLMs, including hallucination, erroneous outputs, and outdated knowledge [5, 6, 7]. To address these limitations, model editing has emerged as a promising approach to precisely update targeted knowledge while preserving the model's general capabilities.

---

*Equal contribution
†Corresponding author

39th Conference on Neural Information Processing Systems (NeurIPS 2025).

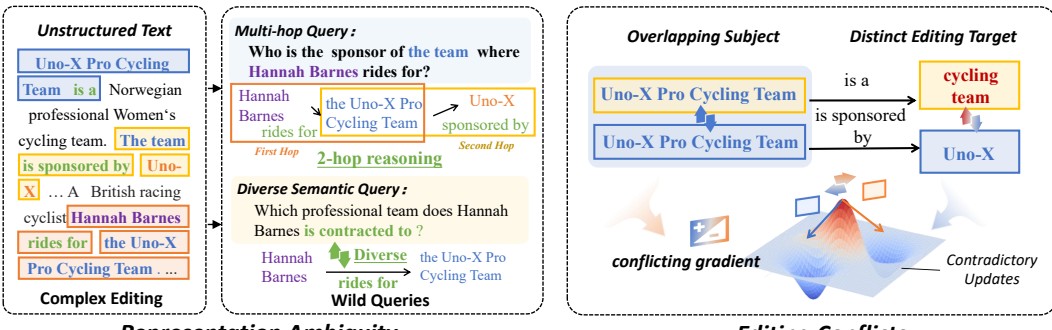

Figure 1: Challenges of WUKE. (1) Representation Ambiguity: The intrinsic complexity of informational interdependencies generates significant uncertainty in knowledge encoding. (2) Editing Conflicts: While structured editing operates on independent samples with low correlation, unstructured text exhibits dense semantic overlaps.

Existing model editing methods predominantly rely on structured knowledge representations, such as triples (*subject, relation, object*) or Question-Answer (QA) pairs. For example, ROME [8] leverages causal mediation analysis over knowledge triples to identify editable regions in Transformer layers, subsequently updating parameters via key-value modifications. MEMIT [9] extends this framework to enable batch editing of multiple triples. In contrast, GRACE [10] maintains a discrete codebook as external memory to track edited knowledge. Recent work like WISE [7] introduces a dual-parameter memory architecture guided by knowledge triples to isolate edits, while AnyEdit [11] employs recursive optimization based on QA pairs to handle variable-length knowledge formats.

However, real-world knowledge is primarily stored in unstructured text (e.g., documents, articles) without explicit question-answer pairs, and transforming such data into structured formats requires costly human annotation, posing significant challenges for knowledge editing in practical applications. To address this, [12] proposed the Wild Unstructured Knowledge Editing (WUKE) paradigm, which aims to extract knowledge from complex unstructured text to update LLMs. Building on WUKE, [12] introduced the AKEW benchmark, which uses raw text passages exclusively for training editing models, while employing knowledge-intensive QA pairs for evaluation. A substantial semantic gap exists between test queries and the edited knowledge—such as semantic paraphrases or multi-hop reasoning over multiple facts—adding difficulty to alignment and evaluation. [12] leveraged large language models to extract knowledge triples from unstructured text, thereby enabling the application of existing structured knowledge editing techniques like ROME[8]. Empirical findings reveal substantial performance deterioration in current methodologies when processing authentic unstructured text, highlighting a crucial unresolved challenge in knowledge editing research.

As illustrated in Figure 2, wild unstructured text manifests complex knowledge entanglement, with real-world queries frequently displaying substantial semantic deviations from source passages, posing persistent challenges for WUKE: (1) **Representation Ambiguity**: The intrinsic complexity of informational interdependencies generates significant uncertainty in knowledge encoding. Contrasted with semantically explicit structured triples, unstructured text interlinks diverse entities through intricate relational networks where single entity may associate with multiple counterparts via distinct relations. Moreover, entities can implicitly connect through transitive relational pathways without direct relations. This complexity is further exacerbated in evaluation scenarios with diverse real-world queries which require robust understanding of unstructured contexts. (2) **Editing Conflicts**: While structured editing operates on independent samples with low correlation, unstructured text exhibits dense semantic overlaps (e.g., shared entities/similar relations). The autoregressive training paradigm exacerbates this issue: concurrent editing of semantically adjacent yet prediction-conflicting statements (e.g., "Paris is in France" versus "Parisian culture emphasizes fashion") induces adversarial gradient directions during parameter optimization. This mechanistic conflict corroborates prior research [13, 14] on editing identical subject entities, where opposing objectives destabilize model convergence, ultimately degrading performance through competing parameter updates.

In this work, we propose a Conflict-Aware Knowledge Editing in the Wild (CAKE) framework for wild unstructured text, which comprises a Semantic-augmented Graph Representation (SGR) and a Conflict-aware Knowledge Editing strategy (CKE). To address knowledge representation ambi-

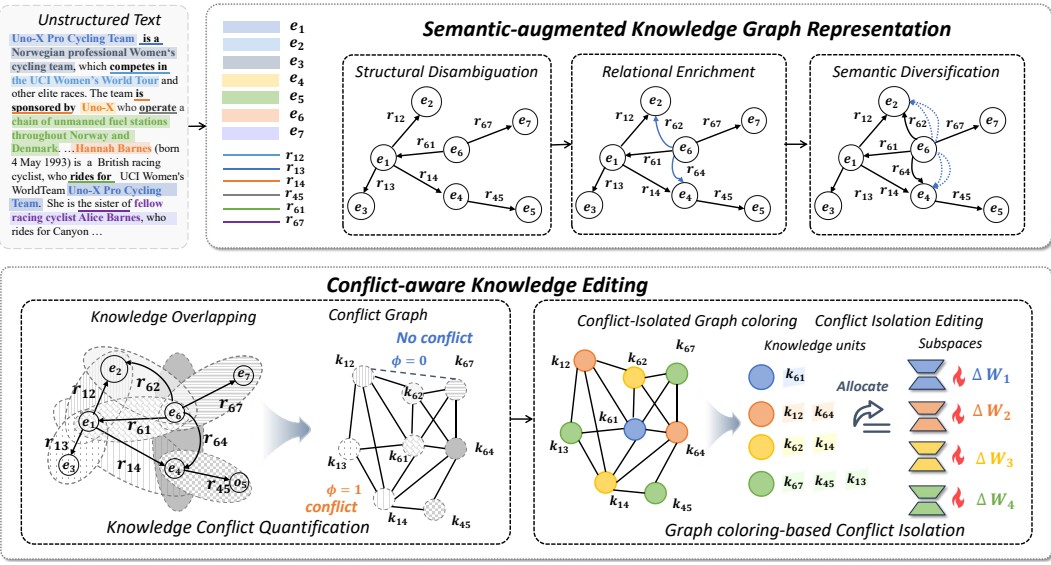

Figure 2: An overview of CAKE. The SGR module first enhances knowledge semantics through three synergistic mechanisms: structural disambiguation, relational enrichment, and semantic diversification. The CKE module introduces a graph-theoretic coloring mechanism that decouples semantically overlapping edits into orthogonal parameter subspaces to eliminate the editing conflicts.

guity, SGR systematically enhances knowledge semantics through three synergistic mechanisms: (1) Structural Disambiguation via dynamic knowledge graph construction from extracted triples, explicitly modeling complex entity interdependencies; (2) Relational Enrichment using multi-hop reasoning to capture latent multidimensional associations; (3) Semantic Diversification through relation paraphrasing that generalizes context-aware predicate variations. This tripartite approach ensures robust encoding of unstructured knowledge while preserving its inherent complexity. Furthermore, to mitigate semantic interference, our CKE introduces a graph-theoretic coloring mechanism that decouples semantically overlapping edits into orthogonal parameter subspaces. Building on semantic-augmented knowledge graph embeddings, we first construct a conflict graph quantifying pairwise editing conflicts through subject entity sharing and relational similarity. Conflicting edits are then allocated to mutually orthogonal subspaces via classical graph coloring, effectively neutralizing gradient interference between entangled knowledge units during optimization.

Our contributions are as follows:

- We propose Conflict-Aware Knowledge Editing, the first knowledge editing framework specifically designed for unstructured text. This framework effectively addresses the challenges of representation ambiguity and Editing conflicts in WUKE, facilitating the application of knowledge editing in real-world scenarios.
- We develop a Semantic-augmented Graph Representation module (SGR) that enhances knowledge semantics through three synergistic mechanisms: structural disambiguation, relational enrichment, and semantic diversification to address the knowledge ambiguity.
- We introduce the Conflict-aware Knowledge Editing strategy (CKE), which introduces a graph-theoretic coloring mechanism that decouples semantically overlapping edits into orthogonal parameter subspaces to eliminate the editing conflicts.
- Extensive experiments conducted on the standard WUKE benchmark demonstrate the effectiveness of our method. In particular, our approach achieves a 15.43% improvement in accuracy over the second-best method on editing llama3.

## 2 Method

Prior knowledge editing methods primarily rely on structured samples and encounter significant challenges when applied to unstructured text due to the inherent complexity of knowledge. These challenges manifest as representation ambiguity and semantic overlap, which constrain the effectiveness of editing. To address these issues, this paper proposes a Conflict-Aware Knowledge Editing

framework for unstructured text in the wild (CAKE). The framework comprises two key components: (1) a Semantic-augmented Graph Representation module (SGR) (Section 2.2), and (2) a Conflict-aware Knowledge Editing (CKE) strategy (Section 2.3).

## 2.1 Preliminary: Wild Unstructured Knowledge Editing

Traditional knowledge editing methods rely on structured knowledge instances $\mathbf{K} = \{\mathbf{k}_i\} = \{(\mathbf{s}_i, \mathbf{r}_i, \mathbf{o}_i)\}$, typically represented as triples, to perform model updates. $\mathbf{s}_i$ and $\mathbf{o}_i$ denote the subject and object entities and $\mathbf{r}_i$ is the relation. During the editing training, the explicit input query is composed of the subject entity and the relation, i.e., $\mathbf{x}_i = \mathbf{s}_i \oplus \mathbf{r}_i$, while the object entity provides supervisory signals as the prediction target.

In contrast, Wild Unstructured Knowledge Editing (WUKE) task [12] necessitates updating language models using solely unstructured textual data — devoid of explicit query-target training pairs — while evaluating model fidelity through wild complex queries that reflect real-world linguistic diversity. A straightforward solution involves extracting discrete knowledge triples from unstructured text and naively replicating the structured editing paradigm by constructing artificial query-target pairs.

However, this adaptation fundamentally misaligns with the nature of unstructured knowledge: whereas structured editing operates on isolated triples $\mathbf{k}_i$ during the whole training phase, WUKE requires single-edit injection of multiple interconnected triples extracted from cohesive textual contexts in only one edit. To clarify the symbolic distinction, we define $\mathbf{k}_{ij} = (\mathbf{e}_i, \mathbf{r}_{ij}, \mathbf{e}_j)$ for WUKE's interconnected triples contrasting with structured editing's knowledge atomic $\mathbf{K} = \{\mathbf{k}_i\} = \{(\mathbf{s}_i, \mathbf{r}_i, \mathbf{o}_i)\}$. $\mathbf{e}_i$ and $\mathbf{e}_j$ denote the subject and object entities and $\mathbf{r}_{ij}$ is the relation. Corresponding pseudo-queries $\mathbf{x}_{ij} = \mathbf{e}_i \oplus \mathbf{r}_{ij}$ could also be analogously constructed, yet ensuring query-answer training.

We adopt the classical parameter-preserving methods as [15], which preserve the original model performance more effectively by introducing additional parameters. Given a pretrained model $\mathcal{M}$ with layer weights $\mathbf{W}$, an unstructured text $\mathbf{T}$, and the extracted knowledge triples $\mathbf{K} = \{\mathbf{k}_{ij}\}$, parameter-preserving methods augment $\mathbf{W}$ with extra parameters $\Delta\mathbf{W}$. The forward pass of the editing layer of the input query $\mathbf{x}_{ij}$ can be denoted as $\mathbf{h}_{ij} = \mathbf{W}\mathbf{a}_{ij} + \Delta\mathbf{W}\mathbf{a}_{ij}$, where $\mathbf{a}_{ij}$ is the representation of $\mathbf{x}$ before feed into layer $\mathbf{W}$. The editing process can be expressed as:

$$\Delta\mathbf{W} \leftarrow \Delta\mathbf{W} - \eta \cdot \left( \frac{1}{|\mathbf{K}|} \sum_{ij} \nabla\mathcal{L}_{CE} \left(\mathbf{e}_j, \mathcal{M}_{t-1}(\mathbf{x}_{ij})\right) \right) \tag{1}$$

where $\eta$ is the learning rate, $\mathcal{M}_{t-1}$ denotes the model after editing step $t-1$ with the additional parameters, $\mathcal{L}_{CE}$ is the cross-entropy loss.

In this paper, we propose the Conflict-Aware Knowledge Editing (CAKE) framework, which comprises two key components: a Semantic-augmented Graph Representation module to mitigate representation ambiguity, and a Conflict-aware Knowledge Editing strategy to resolve the editing conflicts caused by semantic overlap.

## 2.2 Semantic-augmented Graph Representation

The inherent semantic entanglement and complex interdependencies in unstructured text lead to ambiguous knowledge representations under conventional structure-based methods. To address this, we propose the Semantic-augmented Graph Representation (SGR) module, which holistically enhances knowledge encoding through synergistic structural, relational, and diversity-aware semantic reinforcement, optimizing representational granularity and preserving contextual integrity.

**Graph-based Structural Disambiguation**   To holistically capture the latent topological configurations of textual knowledge, we perform dynamic graph induction to establish context-preserving knowledge structures. Unlike discrete triple extractions that fragment relational semantics, our graph-constrained representation paradigm systematically preserves inter-knowledge associations at paragraph-level granularity, maintaining ecological knowledge coherence through structural-semantic alignment. Specifically, we first design a structure-aware prompt template that steers the foundation model $\mathcal{M}$ to dynamically construct a knowledge graph from input text $\mathbf{T}$, ensuring isomorphic mapping between the unstructured text and their underlying knowledge topology:

$$\mathbf{G} = \{\mathbf{E}, \mathbf{R}^s\} = \mathcal{M}([\mathbf{I}^s, \mathbf{T}]) \tag{2}$$

where $\mathbf{E} = \{\mathbf{e}_1, \mathbf{e}_2, \cdots, \mathbf{e}_n\}$ denotes the entity set. $\mathbf{R}^s = \{\mathbf{r}_{ij}\}$ represents the direct relation set. Here $\mathbf{r}_{ij}$ denotes the direct relation between entity $\mathbf{e}_i$ and $\mathbf{e}_j$ lexically instantiated in $\mathbf{T}$, yielding a sparse graph structure. The prompt template $\mathbf{I}^s$ begins with the instruction: "*Extract an interconnected knowledge graph from the paragraph, emphasizing semantic relationships among the extracted triples.*" Further details are provided in the Appendix.

**Multi-hop Relational Enrichment**   While $\mathbf{G}$ encodes the explicit relations among entities in unstructured text $\mathbf{T}$, its latent relational topology remains discontinuous. Establishing multi-hop relational inference pathways between entity pairs enhances comprehension of intricate semantics, enabling robust knowledge editing.

To enhance the semantic continuity, we further guide language model $\mathcal{M}$ to infer latent dependencies between distantly connected entities through $p$-hop dependencies:

$$\mathbf{r}_{ij} = \mathcal{M}([\mathbf{I}^h, \mathbf{G}, \mathbf{T}, (\mathbf{e}_i, \mathbf{e}_j)]), \text{where } d_{\mathbf{G}}(\mathbf{e}_i, \mathbf{e}_j) \in [2, p] \tag{3}$$
$$\mathbf{R}^s \leftarrow \mathbf{R}^s \cup \{\mathbf{r}_{ij}\} \tag{4}$$

Where $\mathbf{I}^h$ denotes the multi-hop reasoning instruction (begin with "*Given a knowledge graph extracted from a text and two nodes, infer and describe the relationship connecting these two nodes.*"), and $d_{\mathbf{G}}$ indicates the path length between entities in $\mathbf{G}$. The augmented relations $\mathbf{R}^s$ allows cross-hop semantic bridges, effectively densifying the knowledge graph's relational manifold while maintaining textual grounding constraints.

**Paraphrase-driven Semantic Diversification**   To address generalization challenges in unstructured editing scenarios, we implement Paraphrase-driven Semantic Diversification, enhancing knowledge representation diversity through controlled lexical variation. For each relational fact $(\mathbf{e}_i, \mathbf{r}_{ij}, \mathbf{e}_j)$ from constructed knowledge($\mathbf{r}_{ij} \in \mathbf{R}^s$), we guide the model $\mathcal{M}$ through a lexical diversification instruction $\mathbf{I}^d$ to generate paraphrastic variants $\mathbf{R}^d$ to augment the semantic diversity:

$$\{\mathbf{r}'_{ij}\} = \mathcal{M}([\mathbf{I}^d, \mathbf{G}, \mathbf{T}, (\mathbf{e}_i, \mathbf{r}_{ij}, \mathbf{e}_j)]), \text{where } \mathbf{r}_{ij} \in \mathbf{R}^s, \tag{5}$$
$$\mathbf{R}^d \leftarrow \mathbf{R}^d \cup \{\mathbf{r}'_{ij}\} \tag{6}$$

The instruction template $\mathbf{I}^d$ begins with "*Rewrite each relation between connected nodes with a semantically equivalent alternative.*") Crucially, the paraphrase-driven diversification operation establishes a bijective mapping between each original relation $\mathbf{r}_{ij} \in \mathbf{R}^s$ and its generated variant set $\{\mathbf{r}'_{ij}\}$, effectively transforming the initial simple graph $\mathbf{G}$ into a directed multigraph $\mathbf{G}' = (\mathbf{E}, \mathbf{R}^s \cup \mathbf{R}^d)$ with multiple parallel edges between entity pairs. This iterative augmentation of edge multiplicity enhances the graph's capacity to encode divergent realizations while preserving semantic invariants, thereby improving robustness against distributional shifts in real-world queries.

The tripartite semantic enhancement framework systematically captures multifaceted relational semantics between entities through structural disambiguation, relation enrichment, and semantic diversification. This synergistic integration dynamically densifies the knowledge representation space derived from unstructured text, establishing comprehensive knowledge manifolds that holistically encode both explicit assertions and latent dependenciesfacilitating the precise knowledge editing.

## 2.3   Conflict-aware Knowledge Editing

Distinct knowledge units extracted from unstructured text exhibit dense semantic overlaps, which induces conflicting editing trajectories. Unlike conventional structured knowledge editing paradigms that process weakly correlated knowledge instances, unstructured text editing confronts semantic overlap phenomena——instances where lexically congruent inputs demand divergent knowledge updates. During gradient-based optimization, these semantically equivalent but knowledge-divergent samples induce conflicting gradient directions.

Therefore, we propose a Conflict-aware Knowledge Editing (CKE) strategy, which formulates the conflicted editing as a graph-coloring problem. CKE first derive a knowledge conflict graph based on our semantic-augmented knowledge graph, and resolves conflicts by assigning distinct colors to conflicting knowledge triples, with each color mapping to an isolated parameter editing subspace.

Specifically, the conflict graph $\mathbf{G}_c^* = (\mathbf{K}, \Phi)$ is constructed, where each vertex $\mathbf{k}_{ij}$ represents a knowledge instance $(\mathbf{e}_i, \mathbf{r}_{ij}, \mathbf{e}_j)$. The edge function $\phi : \mathbf{K} \times \mathbf{K} \to \{0, 1\}$ is defined as:

$$\phi(\mathbf{k}_{ij}, \mathbf{k}_{pq}) = \begin{cases} 1 & \text{if } \alpha \cdot \mathbb{I}(\mathbf{e}_i, \mathbf{e}_p) + \beta \cdot \frac{|\mathbf{a}_{ij}^\top \mathbf{a}_{pq}|}{\|\mathbf{a}_{ij}\| \cdot \|\mathbf{a}_{pq}\|} \geq \gamma \\ 0 & \text{otherwise} \end{cases} \tag{7}$$

where $\mathbf{a}_{ij}$ denotes the hidden representation of $\mathbf{x}_{ij} = \mathbf{e}_i \oplus \mathbf{r}_{ij}$ at the editing layer and $\mathbb{I}(\cdot)$ is the Kronecker delta indicator function. Hyperparameters $\alpha, \beta \in [0, 1]$ control the weights of the subject entity sharing and the query(entity and relation) similarity, respectively, and $\gamma$ serves as the conflict determination threshold.

It is noteworthy that our conflict detection specifically targets knowledge triples with divergent object entities while maintaining shared subject entities or semantically similar relations. For parallel edges in $\mathbf{G}'$ (i.e., triples $(\mathbf{e}_i, \mathbf{r}_{ij}, \mathbf{e}_j)$ and $(\mathbf{e}_i, \mathbf{r}'_{ij}, \mathbf{e}_j)$ with identical subject-object pairs), we explicitly designate them as conflict-free instances and allocate them to the same editing expert. This design rationale explains why Equation 7 focuses solely on the relation component $\mathbf{r}_i j$ and subject entity, as subject-relation pairs constitute the primary conflict determinant while object entities serve as resolution targets.

Following the construction of the knowledge conflict graph, we employ the Welsh-Powell algorithm to compute the vertex coloring scheme $f$ and determine the minimum chromatic number $L$, where $f(\mathbf{k}_{ij}) = \mathbf{c}_l$ assigns the knowledge instance $\mathbf{k}_{ij}$ to the $l$-th color. We initialize a set of LoRA [15] parameter subspaces $\Delta \mathbf{W}_1, ..., \Delta \mathbf{W}_L$ with each subspace updated via gradient descent:

$$\Delta \mathbf{W}_l \leftarrow \Delta \mathbf{W}_l - \eta \cdot \nabla \sum_{f(\mathbf{k}_{ij}) = \mathbf{c}_l} \mathcal{L}\left(\mathcal{M}\left(\mathbf{x}_{ij}\right), \mathbf{e}_j\right) \tag{8}$$

where $\eta$ denotes the learning rate. During post-edit inference, due to the parameter independence among distinct LoRA experts, we aggregate their forward outputs via averaging. To effectively distinguish edited knowledge from the model's original knowledge during inference, we maintain a dynamically preserved knowledge activation vector serving as a memory routing mechanism. Specifically, this memory router computes the similarity between the input activation and stored activation memories; if the similarity falls within a defined threshold, the input is considered within the edit scope. In such cases, outputs are generated by integrating all LoRA subspace parameters along with the original model parameters; otherwise, output relies solely on the original model parameters.

Through our SGR module and CKE strategy, we effectively enhance ambiguous semantics in unstructured texts while mitigating editing conflicts arising from semantic overlaps across heterogeneous knowledge components. This framework significantly improves knowledge editing performance in wild, unstructured textual environments.

## 3 Experiments and Results

### 3.1 Experimental Settings

**Datasets and Evaluation.** Following the standard experimental settings of the WUKE task [12], we evaluate the effectiveness of our method on unstructured text editing using three datasets: CounterFact [8], WikiUpdate [12], and MQuAKE-CF [16]. Consistent with [12], the unstructured text in these datasets is derived from the expansion of specific structured QA pairs. Among them, the CounterFact dataset contains a large amount of counterfactual knowledge, exhibiting a high degree of knowledge conflict within the text. The WikiUpdate dataset features individual texts that encompass a richer and more diverse range of knowledge entities, while the MQuAKE-CF dataset includes more multi-hop relationships among different knowledge points within each text. We assess the editing performance by measuring the accuracy of the model's responses to specific questions after editing. Two key metrics are employed: Reliability (Rel.), which measures the accuracy of the model's answers to questions related to the edited unstructured text, and Locality (Loc.), which evaluates the model's accuracy on questions unrelated to the edited content after the knowledge editing process. For the locality evaluation, we use QA pairs from the ZsRE dataset [17] that are semantically unrelated to the edited samples. Since the WUKE task evaluates editing reliability through abstract questions derived from unstructured text, this reliability metric in fact simultaneously assesses both reliability and generality as defined in conventional structured knowledge editing.

Table 1: Editing Reliability and Locality. Other comparison methods follow the test method set in AKEW. The test samples of Locality are selected from the samples in the ZsRE dataset that are semantically irrelevant to the edited samples for testing.

| Model and Editing Method | | Datasets | | | | | | | | |
|---|---|---|---|---|---|---|---|---|---|---|
| Model | Editing Method | CounterFact | | | MQuAKE-CF | | | WikiUpdate | | |
| | | Rel.(%) | Loc.(%) | Avg(%) | Rel.(%) | Loc.(%) | Avg(%) | Rel.(%) | Loc.(%) | Avg(%) |
| GPT2-XL | ROME | 7.84 | 61.21 | 34.52 | 34.84 | 59.86 | 47.35 | 30.13 | 61.11 | 45.62 |
| | MELO | 1.37 | **64.10** | 32.74 | 19.14 | **63.15** | 41.15 | 29.15 | **63.24** | 46.20 |
| | MEMIT | 5.69 | 59.04 | 32.37 | 33.90 | 60.28 | 47.09 | 34.56 | 62.29 | 48.43 |
| | WISE | 10.01 | 24.16 | 17.09 | 28.58 | 23.10 | 25.84 | 31.27 | 25.33 | 28.30 |
| | Elder | 23.10 | 60.67 | 41.89 | 38.28 | 57.33 | 47.81 | 25.31 | 62.20 | 43.76 |
| | Ours | **35.04** | 61.40 | **48.22** | **62.01** | 61.49 | **61.75** | **42.26** | 62.38 | **51.80** |
| Llama3.2-3B | ROME | 11.56 | 81.28 | 46.42 | 39.56 | **82.74** | 61.15 | 42.43 | **80.68** | 61.56 |
| | MELO | 4.61 | 73.30 | 38.96 | 32.73 | 80.27 | 56.50 | 45.87 | 70.40 | 58.14 |
| | MEMIT | 15.41 | 80.16 | 47.79 | 42.15 | 81.65 | 61.90 | 47.80 | 75.14 | 61.47 |
| | WISE | 4.46 | 14.84 | 9.65 | 19.61 | 11.31 | 15.46 | 4.09 | 17.65 | 10.87 |
| | Elder | 26.04 | 79.23 | 52.64 | 49.19 | 80.44 | 64.82 | 35.40 | 72.30 | 53.85 |
| | Ours | **43.24** | **83.67** | **63.46** | **64.23** | 80.53 | **73.12** | **53.10** | 71.42 | **62.26** |
| Llama3-8b | ROME | 11.64 | 87.32 | 49.48 | 43.54 | 87.44 | 65.49 | 53.11 | 87.87 | 70.49 |
| | MELO | 3.43 | 87.14 | 45.29 | 34.04 | 86.31 | 60.18 | 57.71 | 86.45 | 72.08 |
| | MEMIT | 22.76 | **87.69** | 55.23 | 50.86 | **87.71** | 69.29 | 56.54 | **88.38** | **72.46** |
| | WISE | 21.51 | 44.84 | 33.18 | 47.40 | 43.10 | 45.25 | 28.98 | 51.41 | 40.20 |
| | Elder | 1.99 | 87.27 | 44.63 | 56.13 | 81.50 | 68.82 | **58.23** | 84.35 | 71.29 |
| | Ours | **38.19** | 86.61 | **62.40** | **57.74** | 86.42 | **72.08** | 51.44 | 87.57 | 69.51 |

**Baselines.** We select representative knowledge editing approaches for comparison, including ROME [8], MEMIT [9], WISE [7], Melo [18], and Elder [19]. Since these methods cannot directly perform edits using unstructured text, we follow the settings of AKEW [12] by extracting a corresponding triple from each sentence in the unstructured text and employing these triples as inputs for the editing process in the respective methods.

**Implementation Details .** We validate the effectiveness of the proposed method on three widely adopted large language models (LLMs): GPT2-XL [20], LLaMA3.2-3B [21], and LLaMA3-8B [21]. The default hyperparameter settings are $\alpha = 0.8$, $\beta = 0.8$, and $\gamma = 0.8$ (see Formula 7 in Section 2.3). All experiments are conducted on NVIDIA GPUs, and the editing optimizer is SGD. More experimental details are outlined in the Appendix 5.

## 3.2 Main Results

Table 1 presents the comparative editing performance of our approach and several mainstream knowledge editing methods. Our method CAKE demonstrates superior editing reliability on all three unstructured datasets and achieves comparable results with existing methods in terms of editing locality. The stable performance across different datasets highlights the generalizability of our approach to various types of unstructured texts.

**Performance on the CounterFact dataset.** CAKE achieves outstanding results on the CounterFact dataset, outperforming all existing methods across the three LLMs. Specifically, it surpasses the second-best approach by 11.94%, 17.20%, and 15.43% in reliability. This advantage arises because the CounterFact dataset features significant conflicts between knowledge items and substantial semantic overlap. To address this, our method introduces a graph-coloring-based conflict resolution strategy, which effectively mitigates parameter update inconsistencies caused by knowledge conflicts during editing. Additionally, the incorporated Paraphrase-driven Semantic Diversification mechanism enriches the semantic representations of identical knowledge, substantially improving the editing outcomes for CounterFact samples where test queries differ considerably from the textual triples.

**Performance on the MQuAKE-CF dataset.** On the MQuAKE-CF dataset, CAKE surpasses the reliability of the second-best baseline by more than 15% across all three LLMs. This remarkable improvement can be attributed to the semantic enhancement and conflict resolution mechanisms embedded in CAKE. The MQuAKE-CF dataset presents a higher level of conflict and editing difficulty due to its inclusion of more intricate reasoning dependencies among knowledge entities, such as multi-hop relationships. Our method effectively exploits these implicit relationships through

Table 2: Ablation Study Results. (Reliability % on each Datasets) Avg. represents the mean reliability of the three datasets in the corresponding editing methods and models.

| Model | Ablation Method | Avg. | CounterFact | MQuAKE-CF | WikiUpdate |
|---|---|---|---|---|---|
| GPT2-XL | without SGR | 45.94 | 34.63 | **62.19** | 41.22 |
| | without CKE | 44.13 | 30.06 | 61.51 | 40.84 |
| | Router Expert | 28.89 | 23.10 | 38.28 | 25.31 |
| | Ours | **46.10** | **35.04** | 62.01 | **42.26** |
| Llama3.2-3B | without SGR | 48.63 | 38.38 | 58.94 | 48.56 |
| | without CKE | 50.46 | 34.70 | **65.71** | 50.77 |
| | Router Expert | 36.87 | 26.04 | 49.19 | 35.40 |
| | Ours | **53.52** | **43.24** | 64.23 | **53.10** |
| Llama3-8b | without SGR | 47.62 | 37.28 | 55.53 | 50.06 |
| | without CKE | 47.59 | 25.66 | 56.13 | **60.99** |
| | Router Expert | 30.98 | 1.99 | 32.72 | 58.23 |
| | Ours | **49.12** | **38.19** | **57.74** | 51.44 |

knowledge-graph-based multi-hop reasoning, enabling the model to comprehensively capture and utilize the complex semantic relations inherent in the original text.

**Performance on the WikiUpdate dataset.** On the WikiUpdate dataset, CAKE demonstrates superior editing reliability, exceeding the second-best approach by 7.7% on GPT2-XL. This improvement arises from the dataset's rich inclusion of diverse knowledge entities, where conventional triple extraction techniques often capture only sentence-level associations and fail to account for semantic ambiguities inherent in unstructured text. Our method effectively addresses this limitation through Multi-hop Relational Enrichment, which enhances the detection of inter-entity associations across the entire text and identifies additional knowledge pairs, thereby improving editing reliability. In terms of locality across all models, our approach remains comparable to other state-of-the-art methods.

## 3.3 Ablation Study

We conducted module ablation experiments on all settings as in the main comparative study, comparing the editing reliability before and after module removal to evaluate the effectiveness of the two key components in our method: SGR and CKE. For the ablation of the SGR module, we removed the knowledge graph construction and semantic enhancement processes from the original method. Instead, we adopted the knowledge extraction approach from AKEW to convert unstructured text into a set of triples, while retaining the CKE module for knowledge editing. As shown in Table 2, the editing reliability of the three LLMs decreased by 0.16%, 4.89%, and 1.5%, respectively, compared with the full model. This demonstrates the crucial role of SGR in enabling effective knowledge extraction and semantic enhancement from unstructured text.

For the ablation of the CKE module, two experiments were designed. First, in the without CKE setting, all knowledge representations enhanced by the SGR module were fed into a single LoRA expert for editing. Across the three LLMs, this configuration resulted in a reduction in average editing reliability by 1.97%, 3.06%, and 1.53%, respectively, compared to the complete method. Second, to evaluate the expert allocation effectiveness of CKE, we replaced the graph-coloring-based expert allocation mechanism with a gated routing–based LoRA expert allocation approach, following the design of Elder. This substitution caused an average decrease of over 15% in editing reliability across all three datasets and models. Collectively, these ablation results validate the effectiveness of both the semantic-augmented representation within SGR and the graph-coloring-based expert allocation mechanisms within CKE, highlighting their importance in conflict resolution and in enhancing editing reliability under unstructured text scenarios.

## 3.4 More Analysis

**Analysis of Conflict-aware Hyperparameters.** In order to explore the impact of the hyperparameters controlling the weights of subject entity sharing and query (entity and relationship) similarity in the formula 7 in Section 2.3 for conflict detection of knowledge instances in the CKE module on

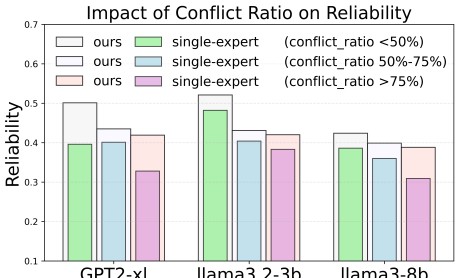 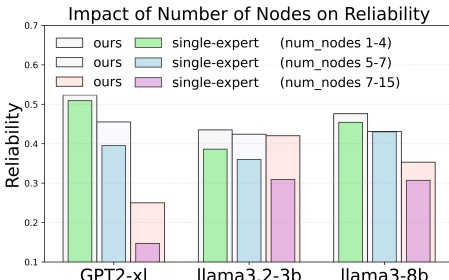

Figure 4: Analysis of the number of nodes in knowledge graph and conflict ratio between knowledge on the reliability of knowledge editing.The average Reliability test results on the three datasets CounterFact/WikiUpdate/MQuAKE-CF.

editing reliability, we observe the impact of different parameter weights on the results by adjusting the values of $\alpha/\gamma$ and $\beta/\gamma$ in the formula. The experimental results are shown in the figure below:

As can be seen from the Fig. 3, the settings of subject entity sharing and the query (entity and relation) similarity in the conflict detection function of the CKE module are reasonable. Both can reflect the conflict of different knowledge in the editing process and will have an impact on editing reliability. A higher similarity detection weight will cause two more similar knowledge examples to be assigned to different experts, thereby improving editing reliability. And as the corresponding weight in the conflict detection function increases, the separation effect of conflicting knowledge will be better.

**Analysis of Knowledge Conflict and Editing Reliability.** To further investigate the relationship among the volume of knowledge, the degree of knowledge conflict, and editing reliability within the knowledge graph, we quantified the degree of knowledge conflict in the CKE algorithm. Specifically, it is defined as the ratio between the number of allocated LoRA experts and the total number of knowledge items. As illustrated in Figure 4, the experimental results demonstrate that as the amount of knowledge increases, the editing reliability generally declines. This observation suggests that the scale of knowledge contained in unstructured text impacts the overall editing performance. However, compared with the single-expert LoRA editing approach, our method exhibits a noticeably slower degradation trend, achieving consistently higher reliability across all knowledge scales.

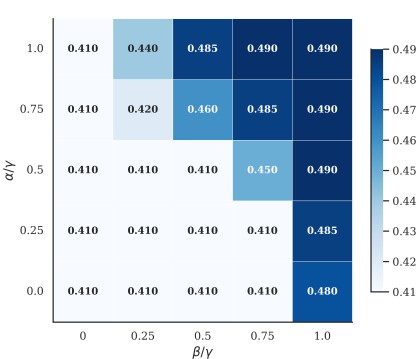

Figure 3: Analysis of Conflict-aware Hyperparameters on LLaMA-3.2.

Moreover, the results indicate that as the degree of conflict among knowledge items within the knowledge graph increases, the editing reliability of the single-expert LoRA method decreases markedly. In contrast, our method maintains a relatively stable level of editing performance under varying degrees of knowledge conflict. These findings highlight that the CKE algorithm—based on knowledge graph coloring—effectively mitigates the performance degradation typically caused by increasing knowledge volume and conflict. This robustness arises from the CKE mechanism's capacity to assign potentially conflicting knowledge (i.e., knowledge that may induce opposite parameter update directions) to separate LoRA experts, thereby isolating their respective update processes and ensuring stable and reliable editing outcomes.

## 4 Related Work

**Knowledge Editing.** Current knowledge editing methods typically rely on carefully curated structured facts as input and evaluate model retention by querying the fact prompts used during the editing process. LoRA [15] introduces additional parameters into the model through low-rank adaptation matrices. Melo [18] constructs a knowledge vector database that incorporates counterfactual reasoning, while ELDER [19] integrates multiple LoRAs through a router network trained to establish

smooth associations among data adapters. However, structured facts are often derived from manually designed datasets and fail to reflect the true source of knowledge updates—namely, unstructured text. In the setting proposed by AKEW [12], the model edits knowledge using triples extracted from each sentence of unstructured text and is evaluated with queries that correspond to these unstructured facts. Under this setting, editing accuracy drops significantly. Recent studies, such as UnKE [22] and AnyEdit [11], have proposed methods for editing unstructured answers. However, these approaches primarily focus on scenarios in which a complete long text serves as the answer to a specific question, incorporating preset queries into the editing process. This setup differs fundamentally from that of AKEW, which performs knowledge editing directly on wild, unstructured text without predefined question constraints. To the best of our knowledge, CAKE is the first method specifically designed for the Wild Unstructured Knowledge Editing (WUKE) task.

**Knowledge Extraction and Modeling.** Knowledge Extraction and Modeling methods can be broadly categorized into two types: traditional methods and those leveraging Large Language Models. Traditional methods for unstructured text knowledge extraction include pipeline and joint models. Pipeline approaches—ranging from rule-based OpenIE[23] to machine learning (e.g., SVM[24]) and deep learning models like BiLSTM-CRF[25], CNN/RNN[26]—suffer from error propagation and limited entity interaction. Joint models such as CopyMTL[27], TPlinker[28], and MBGAB[29] mitigate cascade errors but struggle to capture implicit semantics and global context due to their reliance on surface patterns.In contrast, large language models enable flexible and context-aware knowledge extraction and representation by leveraging pre-trained world knowledge and generative capabilities. They can integrate scattered information, infer implicit relations, and construct structured knowledge with stronger generalization beyond explicit textual cues[30, 31, 32].

## 5 Conclusion

In this work, we first identify the challenging representation ambiguity and editing conflicts lying in Wild Unstructured Knowledge Editing task, and specifically design a Conflict-Aware Knowledge Editing (CAKE) framework for wild unstructured text. The framework integrates a Semantic-augmented Graph Representation module for precise and enriched knowledge encoding, alongside a Conflict-aware Knowledge Editing strategy that isolates conflicting edits through graph-based parameter subspace allocation. Extensive evaluations on the WUKE benchmark verify that CAKE significantly enhances editing accuracy and stability compared to prior approaches. Our method opens a promising path toward effective and conflict-resilient knowledge editing directly on unstructured text, fostering broader adoption of model editing in real-world scenarios.

**Acknowledgment.** This work was supported by the Fundamental Research Funds for the Central Universities.

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

# Appendix

## A. Limitations and Broader Impacts

The limitation of our method is that it can only be applied to LLMs designed for next-token prediction, which makes it unable to adapt to the knowledge editing of LLMs of all architectures. For knowledge editing tasks with strong professionalism in a specific field, during the graph construction process of SGR, it is necessary to use the LLM corresponding to the specific field to guide the graph construction and ensure the accuracy and professionalism of the knowledge graph construction.

In addition, due to the lack of consideration of the authenticity of unstructured text in the framework, some malicious people may use it to incorrectly edit the model to spread false views such as discrimination and prejudice.We suggest that in the future, more supervision should be introduced on the correctness of all texts and other knowledge sources before editing to ensure the reliability of the edited models in terms of ethics and social responsibility.

## B. Implementation Details

### B.1 Editing Details

In the SGR module, for language models capable of accurately following graph-construction instructions (e.g., Llama3-8B), the editing process is directly performed by the model itself. For models with limited instruction-following and generation capabilities (e.g., GPT2-XL), auxiliary assistance is provided by more capable large language models, such as GPT-4. All training and evaluation procedures are conducted on NVIDIA A800 80GB GPUs. Regarding the editing layer of the model, for GPT2-XL, we select "`transformer.h.36.mlp.c_fc`" and "`transformer.h.37.mlp.c_fc`" for editing. For llama3.2-3B and llama3-8B, we select "`model.layers.13.mlp.down_proj`" for editing. During the editing training phase across all experiments, the learning rate is set to 0.001, and the maximum number of iterations is fixed at 50. For experiments involving LoRA-based methods, the rank of each individual LoRA adapter is set to 8.

### B.2 Detailed Instructions in SGR

To support the Semantic-augmented Graph Representation (SGR) module, we define three instruction formats to guide large language models in constructing and enriching unstructured knowledge graphs. Each instruction is accompanied by an in-context example to clarify its intent and output format, followed by an additional example to illustrate generalization.

### 1. Structure-aware Graph Construction

---

**Instruction of Structure-aware Graph Construction with Example**

**Task:** Extract an interconnected knowledge graph from the following paragraph. Each triple should represent a clear semantic relation between two entities explicitly stated in the paragraph. Preserve the contextual semantics. Output as a list of triples in the form: (Entity1, Relation, Entity2).

**Example Input/Output:**

**Input Paragraph:**
"`Alan Turing developed the concept of a Turing machine, which became the foundation of computer science. He worked at Bletchley Park during World War II.`"

**Output Triples:**
- (Alan Turing, developed, concept of a Turing machine)
- (Turing machine, became the foundation of, computer science)
- (Alan Turing, worked at, Bletchley Park)
- (Bletchley Park, active during, World War II)

---

## 2. Multi-hop Relational Enrichment

**Instruction of Multi-hop Relational Enrichment with Example**

**Task:** Given a knowledge graph extracted from a paragraph and two nodes, infer and describe the semantic relation that connects them indirectly via multi-hop reasoning. Ensure the relation reflects coherent real-world semantics grounded in the original text.

**Example Input/Output:**

**Input Graph:**

- (Marie Curie, discovered, radium)
- (radium, used in, cancer treatment)

**Text:**
"Marie Curie discovered the radioactive element radium, which later found applications in cancer therapy."

**Query:** (Marie Curie, ?, cancer treatment)

**Output:** (Marie Curie, discovered substance used in, cancer treatment)

## 3. Paraphrase-driven Semantic Diversification

**Instruction of Paraphrase-driven Semantic Diversification with Example**

**Task:** Given a knowledge graph with triples, rewrite each relation in a semantically equivalent but lexically different way to diversify the expression of the knowledge. Keep entity names unchanged. Return paraphrased triples.

**Example Input/Output:**

**Input Triple:** (Nikola Tesla, invented, alternating current)

**Output Paraphrases:**

- (Nikola Tesla, was the inventor of, alternating current)
- (Nikola Tesla, came up with, alternating current)
- (Nikola Tesla, pioneered, alternating current)

## B.3 Details of the activation-similarity routing mechanism

In our framework, the activation-similarity routing mechanism is not a central design component, as the underlying principle is relatively conventional. Consequently, it is not emphasized in the main body of the paper. The activation-similarity routing employs a hard activation-based routing strategy to determine whether an input instance falls within the scope of previously edited knowledge. Specifically, during the editing phase, we preserve low-bit lightweight activation vectors corresponding to each edited knowledge item. During inference, the cosine similarity between the input prompt and the stored activation representations is computed. If the similarity exceeds a predefined threshold (set to 0.7 in our experiments), the input is deemed relevant to the edited knowledge, and the corresponding LoRA module is activated for inference. Otherwise, the model relies solely on its original parameters without invoking any edited components.

## C. Descriptions of Compared Model Editors

We compare our method with five representative model editing approaches. Below we summarize their core mechanisms and editing properties.

**ROME** [8]: ROME identifies the internal location of factual associations in LLMs using causal tracing, assuming that MLP layers are the main knowledge carriers. It edits the value matrix of a

single MLP layer using a closed-form rank-one update derived from least squares regression. ROME injects one fact per iteration, enforcing precision through a Lagrangian remainder, but lacks scalability in multi-edit or lifelong scenarios.

**MELO** [18]: MELO assumes that attention heads store factual knowledge and adopts a discrete adapter allocation strategy. It assigns a dedicated LoRA adapter to each knowledge edit, enabling isolation and preventing interference. Although effective in reliability, its discrete mapping is brittle to paraphrased or slightly varied inputs, making it less robust for generalization in lifelong settings.

**MEMIT** [9]: MEMIT builds on ROME's assumptions, treating FFNs as knowledge key-value stores, and extends editing to multiple layers and multiple facts simultaneously. It directly modifies FFN parameters using batched least-squares updates, supporting thousands of edits. However, edits accumulate in the same parameter space, which can cause interference and poor locality during lifelong updates.

**WISE** [7]: WISE introduces a dual parametric memory mechanism to bridge the gap between long-term memory (model parameters) and working memory (retrieval-based representations). It edits only the side memory—a copied FFN value matrix—and routes queries between main and side memory based on an activation-based routing mechanism. To support lifelong editing, WISE uses knowledge sharding into orthogonal subspaces and merges them via Ties-Merge, achieving high reliability, locality, and generalization even with thousands of edits.

**ELDER** [19]: ELDER enhances lifelong model editing by adopting a Mixture-of-LoRA (MoL) approach, replacing discrete adapter selection with a continuous, learnable router that adaptively combines top-$k$ LoRA modules per input. This enables robust handling of semantically equivalent paraphrases. A guided loss aligns edit knowledge with LoRA allocations, while a deferral mechanism detects whether an input needs editing, allowing fallback to the original model for general tasks. ELDER avoids the scalability bottleneck of discrete adapter assignment and maintains strong generalization and reliability under long edit sequences.

## D. More Experiments

### D.1 Editing Efficiency

To evaluate the training efficiency of the proposed editing framework, we compared its per-sample editing time with that of several representative baseline methods. As shown in Table 3, the proposed method demonstrates superior efficiency during the editing process. Compared with post-hoc editing approaches such as ROME and MEMIT, the isolated low-rank parameter training performed by the LoRA mechanism achieves convergence with substantially lower computational cost. Moreover, in comparison with methods such as MELO and Elder, our approach benefits from the coloring strategy based on the Welsh–Powell algorithm in the CKE module, which eliminates the need for additional training. Consequently, it is more efficient than both the expert-selection gating mechanism of Elder, which requires extra training, and the MELO method, which entails extensive vector data access. These results collectively indicate that the proposed method achieves notable advantages in editing efficiency compared with existing model editing approaches.

Table 3: Comparison of average editing time per sample (in seconds).Tested on Llama3-8b from the dataset MQuAKE-CF.

| Method | ROME | MELO | MEMIT | WISE | ELDER | **Ours** |
|---|---|---|---|---|---|---|
| Time (s) | 78.64 | 52.76 | 77.23 | 143.91 | 54.17 | **49.31** |

### D.2 Analysis of threshold in activation-similarity routing

The Figure 5 below illustrates the effect of the similarity threshold in activation-similarity routing on the CounterFact dataset using Llama3.2. As the threshold decreases, editing reliability tends to decline, while locality improves. This trend arises because a lower threshold enforces a stricter

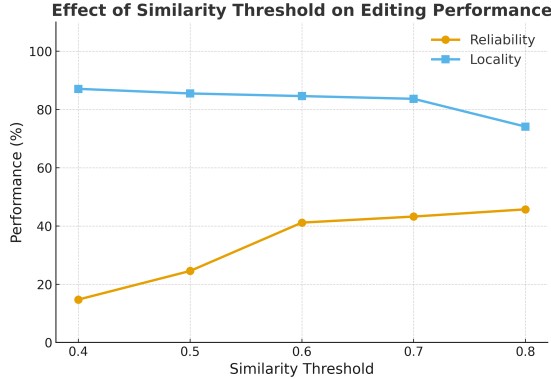

Figure 5: Performance at different similarity thresholds in activation-similarity routing.

criterion for LoRA expert activation, leading the model to depend more heavily on its unedited parameters during inference.

### D.3 Evaluation of the Initial Knowledge Graph Quality in SGR

The quality of the initial knowledge graph is fundamental to CAKE's success. Through manual evaluation, we verified that CAKE's initial knowledge graph accurately and comprehensively covers nearly all knowledge entities present in the original unstructured passage. We report the corresponding evaluation results in the Table 4. We conducted a human evaluation from two perspectives: extraction accuracy and knowledge completeness. As can be seen, CAKE's IKG achieves near-perfect coverage of the entities and relations present in the raw text. This strong performance mainly stems from (i) CAKE's carefully designed, task-specific prompting rules and (ii) the capabilities of the employed LLM (for models with limited intrinsic extraction capacity, such as GPT2-XL, we leverage the GPT-4 API for IKG construction).

Table 4: Evaluation of the Initial Knowledge Graph Quality in SGR.

| **Datasets** | knowledge completeness | extraction accuracy |
|---|---|---|
| CounterFact | 0.99 | 0.95 |
| MQuAKE-CF | 0.99 | 0.97 |
| WikiUpdate | 0.93 | 0.95 |

### D.4 Detialed Ablation of SGR

Table 5: Detialed Ablation of SGR. Each component within the SGR module yields a distinct improvement in editing performance.

| **Method** | w/o SGR | w/o Multi-hop Enrichment | w/o Paraphrasing | CAKE |
|---|---|---|---|---|
| Time (s) | 38.38 | 39.78 | 41.66 | **43.24** |

To further verify that each semantic enhancement component within the SGR module contributes meaningfully to the overall editing performance, we conducted ablation studies on the Multi-hop Relational Enrichment and Paraphrase-driven Semantic Diversification stages. These experiments were designed to examine how representational ambiguity influences the model's editing capability. All evaluations were performed on the CounterFact dataset using the Llama3.2-3B model, and the results are summarized in Table 5. As shown, each component within the SGR module yields a distinct improvement in editing performance, confirming that the proposed Multi-hop Relational Enrichment and Paraphrase-driven Semantic Diversification mechanisms effectively enhance the model's ability to capture diverse relational structures and achieve a more comprehensive representation of factual knowledge contained in unstructured text.

## E. End-to-End Case Study of CAKE Framework

Below I will construct a complete unstructured text knowledge editing process example based on the method framework (**CAKE**, including **SGR** and **CKE**), covering the entire process from:

Step 0: Input Unstructured Text

Step 1: Structure-aware Graph Construction

Step 2: Multi-hop Relational Enrichment

Step 3: Paraphrase-driven Semantic Diversification

Step 4: Conflict Graph Construction

Step 5: Conflict-aware Graph Coloring

Step 6: Final Answer after Editing

---

### Step 0: Input Unstructured Text

**Input Paragraph:**
```
"Marie Curie discovered the radioactive element radium, which was later
used in cancer therapy. She received a Nobel Prize for her contributions.
Radium decays into radon, which is a health hazard. Her husband Pierre
Curie was also a physicist."
```

---

### Step 1: Structure-aware Graph Construction

Using a structure-aware prompt to extract knowledge triples:
**Output Triples:**

- (Marie Curie, discovered, radium)
- (radium, used in, cancer therapy)
- (Marie Curie, received, Nobel Prize)
- (radium, decays into, radon)
- (radon, is, health hazard)
- (Pierre Curie, was, physicist)
- (Pierre Curie, was husband of, Marie Curie)

Resulting in base knowledge graph $\mathbf{G} = \{\mathbf{E}, \mathbf{R}^s\}$.

---

### Step 2: Multi-hop Relational Enrichment

Using prompt-based inference for disconnected entity pairs:
**Example:** For (Marie Curie, cancer therapy)
**Output:** (Marie Curie, discovered radium used in, cancer therapy)
This relation is added to form the enriched graph $\mathbf{G}$.

## Step 3: Paraphrase-driven Semantic Diversification

For each $(e_i, r_{ij}, e_j)$ in $\mathbf{R}^{\text{s}}$, generate lexical variants:
**Input:** (Marie Curie, discovered, radium)
**Paraphrases:**

- (Marie Curie, was the discoverer of, radium)
- (Marie Curie, identified, radium)
- (Marie Curie, found, radium)

The resulting graph becomes a multigraph $\mathbf{G}' = (\mathbf{E}, \mathbf{R}^{\text{s}} \cup \mathbf{R}^{\text{d}})$.

## Step 4: Conflict Graph Construction

Construct the conflict graph $\mathbf{G}_c^* = (\mathbf{K}, \Phi)$ where each node is a knowledge triple $k_{ij} = (\mathbf{e}_i, \mathbf{r}_{\text{ij}}, \mathbf{e}_j)$.

**Example Conflict Detection of $k_{12}$ with triples $k_{13}$, $k_{41}$:**

- $k_{12}$: (Marie Curie, received, Nobel Prize)
- $k_{13}$: (Marie Curie, discovered, radium)
- $k_{41}$: (Pierre Curie, was husband of, Marie Curie)

**Conflict Detection Criterion:** $\phi(\mathbf{k}_{ij}, \mathbf{k}_{pq})$

- $\varphi(k_{12}, k_{13}) = 1$    (high subject or relation semantic similarity)
- $\varphi(k_{12}, k_{41}) = 0$    (low subject or relation semantic similarity)

Hence, in the conflict graph $\mathbf{G}_c^*$, $k_{12}$ has an edge with $k_{13}$, but not with $k_{41}$.

## Step 5: Conflict-aware Graph Coloring

Apply the Welsh-Powell algorithm on $\mathbf{G}_c^*$ to assign edit-isolating colors. For three knowledge triples $k_{12}$, $k_{13}$, $k_{41}$:

**Vertex Coloring Result:**

- **Color 1** (assigned to $\Delta W_1$): $k_{12}$ (Marie Curie, received, Nobel Prize)
- **Color 2** (assigned to $\Delta W_2$): $k_{13}$ (Marie Curie, discovered, radium)
- **Color 1** (shared with $k_{12}$): $k_{41}$ (Pierre Curie, was husband of, Marie Curie)

Note that $k_{12}$ and $k_{41}$ share no conflict and thus share LoRA subspace $\Delta W_1$.

**Update Rule per Color Group:**

$$\Delta \mathbf{W}_l \leftarrow \Delta \mathbf{W}_l - \eta \cdot \nabla \sum_{f(\mathbf{k}_{ij})=\mathbf{c}_l} \mathcal{L}\left(\mathcal{M}\left(\mathbf{x}_{ij}\right), \mathbf{e}_j\right) \tag{9}$$

During optimization:

- Color 1 ($\Delta W_1$) updates $k_{12}$ and $k_{41}$
- Color 2 ($\Delta W_2$) independently updates $k_{13}$ to avoid gradient conflict with $k_{12}$

## Step 6: Final Answer after Editing

At inference time, use all LoRAs to perform inference.

**Query:** "Did Marie Curie receive the Nobel Prize?"
**Edited Model Output:** "Yes, she did."

