# OpenReview forum: "Conflict-Aware Knowledge Editing in the Wild: Semantic-Augmented Graph Representation for Unstructured Text"
_NeurIPS.cc/2025/Conference — NeurIPS 2025 spotlight_

### Official Review · Reviewer_NqoV · 2025-06-23

**Clarity:** 3
**Significance:** 3
**Originality:** 3
**Rating:** 4
**Confidence:** 3

**Summary:**

The authors propose conflict-aware knowledge editing, which is the first knowledge editing framework specifically designed for unstructured text, which effectively addresses the challenges of representation ambiguity and editing conflicts in the wild unstructured text editing.

**Questions:**

1.	The proposed method performs better on MQuAKE-CF after removing CKE. Could this be attributed to MQuAKE-CF's specific features?
2.	In Section 2.2, how exactly are the prompt templates designed, could you list the prompt templates?
3.	Can the conflict definition in Section 2.3 cover all scenarios, such as triples with the same relation and object but subjects are different?

**Ethical Concerns:**

["NO or VERY MINOR ethics concerns only"]

**Final Justification:**

The authors addressed most of the concerns.

**Limitations:**

The description of the experimental setups need to be more detailed, such as the specific design of prompt templates and the selection of hyperparameters in Formula (7), to facilitate readers' understanding.

**Paper Formatting Concerns:**

There are no paper formatting concerns.

**Quality:**

3

**Strengths And Weaknesses:**

Strengths:

1.	To enhance knowledge editing on unstructured text, the authors mitigate knowledge ambiguity via a semantic-enhanced graph representation, and resolve editing conflicts through a conflict-aware knowledge editing strategy based on graph-theoretic coloring.
2.	The experimental results validate the effectiveness of the method proposed by the authors across different datasets and models of varying scales.


Weaknesses:

1.	The three hyperparameters $\alpha$, $\beta$, and $\gamma$ in Formula (7) are all set to a default value of 0.8. However, the paper does not provide any justification or guidance on how these values are chosen or how to determine their optimal settings.
2.	The proposed method did not outperform the baseline on the Loc. metric in most cases, could you provide a detailed analysis of the underlying reasons for this phenomenon?
3.	There are some typos, for example, in Line 127 “can be denotes as” $\rightarrow$ “can be denoted as”.

---

> ### Author Rebuttal · Authors · 2025-07-31
>
> ## Response to Reviewer NqoV
>
> > ***(Weaknesses 1)*** *The three hyperparameters,, andin Formula (7) are all set to a default value of 0.8. However, the paper does not provide any justification or guidance on how these values are chosen or how to determine their optimal settings.*
>
> Thank you for your valuable suggestions. As the values of  ($\alpha, \beta, \gamma$)  vary, CAKE exhibits reasonable performance fluctuations within a certain range, demonstrating overall robustness. We have provide an ablation study on the impact of the hyperparameters $\alpha, \beta, \gamma$ in Appendix D.1 "*Ablation on Conflict-aware Hyperparameters*".
>
> Specifically, we dynamically vary the ratios $\alpha/\gamma$ and $\beta/\gamma$ in Equation (7) and report the corresponding editing accuracy in Figure 4. The results show that increasing $\alpha/\gamma$ and $\beta/\gamma$ lowers the threshold for identifying knowledge conflicts, leading to more knowledge being classified as conflicting and thus isolated into different experts. This enhances editing reliability, albeit with a slight increase in the number of LoRA modules and associated parameters.
>
> > ***(Weaknesses 2)*** *The proposed method did not outperform the baseline on the Loc. metric in most cases, could you provide a detailed analysis of the underlying reasons for this phenomenon?*
>
> Thank you for your valuable suggestions. While achieving a substantial improvement in reliability, CAKE still maintains locality performance comparable to that of existing methods, and its average locality across the three unstructured-text editing datasets surpasses all baselines. The slight degradation observed in a few configurations is likely due to minor routing biases in the memory-routing module during inference. In future work, we will refine this component and explore more effective inference-path selection mechanisms to further boost CAKE’s locality.
>
> > ***(Weaknesses 3)*** *There are some typos, for example, in Line 127 “can be denotes as”“can be denoted as”.*
>
> Thank you for your thoughtful and valuable suggestions. We have revised the corresponding sections accordingly and will further review and refine the entire manuscript in the final version to ensure clarity and readability.
>
> > ***(Question 1)*** *The proposed method performs better on MQuAKE-CF after removing CKE. Could this be attributed to MQuAKE-CF's specific features?*
>
> Thank you for your valuable suggestions. CKE’s relatively limited performance on MQuAKE-CF is attributable to the dataset's lower degree of knowledge conflicts, resulting in only modest editing improvements from CKE. MQuAKE-CF has strong multi-hop reasoning and context-dependent features. The multi-hop enhancement of the SGR module can already resolve knowledge ambiguity very well. Therefore, the contribution of CKE is not significant compared with other datasets.
>
> > ***(Question 2)*** *In Section 2.2, how exactly are the prompt templates designed, could you list the prompt templates?*
>
> Thank you for your valuable suggestions. The structured prompting templates for each module described in Section 2.2 are provided in the Appendix E.
>
> > ***(Question 3)*** *Can the conflict definition in Section 2.3 cover all scenarios, such as triples with the same relation and object but subjects are different?*
>
> Thank you for your valuable suggestions. The conflict definition in Section 2.3 covers cases involving the same relation but does not include cases with the same object. Specifically, as analyzed in Section 2.3, identical relations typically lead to semantically similar input prompts. When different targets are enforced on such similar inputs, conflicting update directions may arise. In contrast, when the objects are the same, the editing objectives are aligned, resulting in consistent update trends and directions, and thus do not constitute a conflict under our definition.

---

### Official Review · Reviewer_Rpyk · 2025-06-29

**Clarity:** 3
**Significance:** 2
**Originality:** 3
**Rating:** 4
**Confidence:** 3

**Summary:**

In this paper, the authors propose the CAKE framework, which tackles the knowledge editing in LLM with natural, unstructured, and possibly concept-conflicted knowledge. Specifically, the CAKE first uses a carefully designed pipeline to extract triplets from the unstructured text. Next, the knowledge edit is conducted by first constructing a conflict graph using the approximated conflict score. Then, the graph coloring algorithm is used to partition the conflict graph into different parts. Finally, each subgraph is edited by training different LoRA. Experiments confirm the effectiveness of the proposed method.

**Questions:**

See the weakness section.

**Ethical Concerns:**

["NO or VERY MINOR ethics concerns only"]

**Final Justification:**

### Justification for not a lower score:
The paper is technically sound, and the writing is clear. The motivation, proposed methods, and results are clear and consistent. The proposed methods show improvement without overwhelming cost.

### Justification for not a higher score.
Overall, the proposed method is not a very solid contribution but more like a moderate improvement.

**Limitations:**

The biggest limitation to me is how the problems tackled by the proposed method actually caused a significant burden to the downstream editing performance. And how is the improvement of introducing the proposed method worth the cost?

**Paper Formatting Concerns:**

Looks good to me.

**Quality:**

2

**Strengths And Weaknesses:**

## Strengths:

- The writing of the paper and the workflow of the proposed method are clear to me.
- The proposed pipeline is effective based on the experimental results. The design of conflict graph construction and graph coloring is interesting.

## Weaknesses:
- The proposed method is based on two motivations: representation ambiguity and editing conflict. If I understand it correctly, the representation ambiguity is solved by the SGR, and the editing conflict is solved by CKE. I have some questions on both parts.
- For representation ambiguity. There is no analysis on how often the problem exists in the real-world data and how such ambiguity will affect the knowledge editing performance. One possible way is that maybe include an ablation study on each submodule in SGR that affects the performance (I only see the ablation study on the whole SGR part). It would be even better to include results on how much the effect is in the raw data and how the SGR mitigates this effect.
- For the editing conflict part, it is also not clear to me how large the effect of that is on the downstream performance.
- For CKE module, I am wondering what the distribution of the results conflict graph (average node, edges), graph partition, and how many LoRA are actually used.  When comparing with a single expert, have authors consider the effect of parameter size?

---

> ### Author Rebuttal · Authors · 2025-07-31
>
> ## Response to Reviewer Rpyk
>
> > ***(Weakness 1 and 2)*** *The proposed method is based on two motivations: representation ambiguity and editing conflict. If I understand it correctly, the representation ambiguity is solved by the SGR, and the editing conflict is solved by CKE. I have some questions on both parts. For representation ambiguity. There is no analysis on how often the problem exists in the real-world data and how such ambiguity will affect the knowledge editing performance. One possible way is that maybe include an ablation study on each submodule in SGR that affects the performance (I only see the ablation study on the whole SGR part). It would be even better to include results on how much the effect is in the raw data and how the SGR mitigates this effect.*
>
> Thank you for your valuable suggestions. Each component of the SGR module contributes to a clear improvement in editing performance, validating both the impact of ambiguity on model behavior and the effectiveness of CAKE in addressing it.  Following your recommendation, we conducted ablation experiments on the “Multi-hop Relational Enrichment” and “Paraphrase-driven Semantic Diversification” steps within our SGR module to investigate how representational ambiguity influences editing performance. We conducted experiments on the CounterFact dataset on llama3.2-3b. The results are presented below：
>
> ||||||
> |-|-|-|-|-|
> ||baseline|w/o Multi-hop|w/o Paraphrase-driven|CAKE|
> |editing reliability|38.38|39.78  |41.66  |43.24|
>
> As shown, each component of the SGR module contributes to a clear improvement in editing performance, further demonstrating that the proposed “Multi-hop Relational Enrichment” and “Paraphrase-driven Semantic Diversification” strategies effectively strengthen diverse types of relational knowledge and enable a more complete modeling of facts expressed in unstructured text.
>
> > ***(Weakness 3)*** *For the editing conflict part, it is also not clear to me how large the effect of that is on the downstream performance.*
>
> Thank you for your valuable suggestions. Our experiments reveal that editing conflicts significantly degrade overall editing performance, whereas our method effectively mitigates this issue. In Table 2 of the paper we report the editing performance obtained (i) after removing the CKE module and (ii) when a random LoRA-assignment strategy is adopted. The results show a clear performance drop when editing conflicts are no longer isolated, which demonstrates that the edited knowledge contains a substantial number of conflicts and that resolving these conflicts with CKE effectively boosts overall editing quality.
>
> Furthermore, in Section 3.4 we present how editing performance varies with the number of knowledge conflicts (see Figure 3, left). As the conflict count increases, overall performance declines slightly, since higher conflict density raises the difficulty of optimization; nevertheless, our method retains a significant margin over the baseline. These observations corroborate the decisive impact of editing conflicts on downstream performance.
>
> > ***(Weakness 4)*** *For CKE module, I am wondering what the distribution of the results conflict graph (average node, edges), graph partition, and how many LoRA are actually used. When comparing with a single expert, have authors consider the effect of parameter size? *
>
> Thank you for your valuable suggestions. We have collected detailed statistics regarding the conflict graphs used in the CKE module, including the average number of nodes and edges per graph, the graph partitioning results, and the actual number of LoRA experts employed, as shown in table below:
>
> | Statistical indicators         | Avg | Main distribution range |
> | ------------------------------ | :-: | :---------------------: |
> | Average number of LoRA         |  5  |           3–10          |
> | Number of conflict graph edges |  12 |           5–20          |
> | Number of conflict graph nodes |  10 |           6–15          |
>
> We will append the whole figure of distribution in the final vision.
>
> Moreover, When compared with a single expert whose total number of parameters is comparable to ours, our method still demonstrates a significant improvement in performance, as shown in table below. This further confirms that subspace assignment via multiple LoRAs effectively mitigates editing conflicts.
>
> | Model & Method |  Avg  | CounterFact | MQuAKE-CF | WIKIupdate |
> | :-------------------- | :---: | :---------: | :-------: | :--------: |
> | **GPT2-xl**           |       |             |           |            |
> |Single LoRA Expert with the same parameters| 44.13 |    30.06    |   61.51   |    40.84   |
> | Ours| 46.10 |    35.04    |   62.01   |    42.26   |
> | **Llama3.2-3b**       |       |             |           |            |
> |Single LoRA Expert with the same parameters| 50.46 |    34.70    |   65.71   |    50.77   |
> | Ours              | 53.52 |    43.24    |   64.23   |    53.10   |
> | **Llama3-8b**         |       |             |           |            |
> |Single LoRA Expert with the same parameters| 47.59 |    25.66    |   56.13   |    60.99   |
> | Ours              | 49.12 |    38.19    |   57.74   |    51.44   |
>
> We have carefully addressed the comments point by point, and we hope these revisions will further meet your expectations and gain your approval.

---

> > ### Comment · Reviewer_Rpyk · 2025-08-04
> > **Thanks**
> >
> > Thank authors for the detailed response. I do not have further questions and will consider increasing my score.

---

> > > ### Author Response · Authors · 2025-08-05
> > >
> > > Thank you for your feedback! We appreciate your constructive reviews for improving our work.

---

### Official Review · Reviewer_R5dq · 2025-06-30

**Clarity:** 2
**Significance:** 2
**Originality:** 2
**Rating:** 4
**Confidence:** 4

**Summary:**

The paper introduces the Conflict-Aware Knowledge Editing in the Wild (CAKE) framework, a novel approach for updating the knowledge of Large Language Models (LLMs) directly from unstructured text. The authors identify two key challenges in this process: representation ambiguity, where the complex relationships in text are hard to encode, and editing conflicts, where updating one piece of information negatively impacts related but distinct facts. The authors evaluated CAKE on the AKEW benchmark using several LLMs, including GPT2-XL, Llama3.2-3b, and Llama3-8b. The experimental results demonstrate that CAKE significantly outperforms existing model editing methods in terms of reliability while maintaining performance on unrelated knowledge.

**Questions:**

The SGR module seems to introduce significant computational overhead due to its reliance on an LLM for graph construction, enrichment, and diversification. Could you provide an analysis of the computational cost (e.g., time, GPU hours) of CAKE compared to the baseline methods?

The quality of the initial knowledge graph is fundamental to CAKE's success. How does the framework handle inaccuracies, such as missed entities or incorrect relations, from the LLM-based graph extraction step? Have you studied how the quality of this initial graph correlates with final editing performance?

The conflict detection mechanism relies on hyperparameters α, β, and γ. How did you arrive at the default value of 0.8 for each? How sensitive is the framework's performance, particularly the trade-off between reliability and locality, to changes in these values?

Could you please elaborate on the implementation of the "memory routing mechanism" used at inference time? Specifically, how are the "activation memories" for edited knowledge created and stored, and what method is used to set the similarity threshold for engaging the LoRA experts?

What is the typical number of "colors" (i.e., LoRA experts) required for the datasets in your experiments? How does the system's performance and parameter count scale as the textual input becomes more complex and the number of knowledge conflicts increases?

Your ablation study shows that using a single expert for all edits is suboptimal.  To further isolate the benefit of your conflict-aware strategy, have you considered comparing the graph-coloring allocation to a simpler baseline, such as randomly assigning conflicting edits to a set of available experts?

**Ethical Concerns:**

["NO or VERY MINOR ethics concerns only"]

**Final Justification:**

The authors addressed my concerns.

**Limitations:**

yes

**Quality:**

2

**Strengths And Weaknesses:**

**strength:**

It tackles the challenge of editing LLMs using real-world, unstructured text, moving beyond the limitation of methods that require pre-structured knowledge triples or question-answer pairs. The CAKE framework is an innovative solution that systematically addresses the core problems of ambiguity and conflict. The two-module design is logical, with each component targeting a specific, well-defined challenge. The paper provides robust empirical evidence of CAKE's effectiveness across multiple models and datasets from the AKEW benchmark. The authors include detailed ablation studies that validate the contribution of each component of the framework.


**weaknesses:**
1. The SGR module requires several interactions with a large language model to construct and enrich the knowledge graph (for extraction, multi-hop reasoning, and paraphrasing).  This process appears computationally intensive, but the paper does not analyze the associated overhead in terms of time or resources, which could be a significant barrier to practical implementation.

2. The entire framework's performance hinges on the ability of a foundational LLM to accurately extract entities and relations to form the initial knowledge graph.  The paper does not sufficiently discuss how errors or biases from this initial step might propagate through the system and affect the final editing performance.

3. The conflict detection formula includes several hyperparameters (α, β, γ) that are crucial for determining whether two pieces of knowledge are in conflict.  The authors provide default values but do not present an analysis of how sensitive the model's performance is to variations in these values, which is important for understanding the framework's robustness.

4. The mechanism for choosing between the original model and the edited parameters during inference is described as a "memory routing mechanism."  However, the paper lacks specific details about how this router is implemented, how activation memories are stored, or how the similarity threshold is determined, making this critical component difficult to reproduce or evaluate.


minor mistake: L208 missing reference of LoRA

---

> ### Author Rebuttal · Authors · 2025-07-31
>
> ## Response to Reviewer R5dq
> We appreciate the reviewer for the detailed comments. We have carefully followed up on and settled every issue raised by them.
>
> > ***(Weakness1)***  *The SGR module requires several interactions with a large language model to construct and enrich the knowledge graph (for extraction, multi-hop reasoning, and paraphrasing). This process appears computationally intensive, but the paper does not analyze the associated overhead in terms of time or resources, which could be a significant barrier to practical implementation.*
>
> > ***(Question 1)***  *The SGR module seems to introduce significant computational overhead due to its reliance on an LLM for graph construction, enrichment, and diversification. Could you provide an analysis of the computational cost (e.g., time, GPU hours) of CAKE compared to the baseline methods?*
>
> Thank you for your valuable suggestions. The extra time incurred by SGR module is minimal for the sparsity of the graph. In the table below, we provide a comparison of editing times between our approach and prior unstructured knowledge-editing methods [1]. We conducted experiments on the CounterFact dataset on llama3.2.
>
> ||||
> |-|-|-|
> ||baseline|CAKE|
> |average time(s)|7.16s|8.10s|
>
> As shown, compared to the baseline, the additional computation time introduced by the SGR module is minimal. This is because, in knowledge editing based on unstructured text, the extraction and modeling of knowledge are indispensable for constructing the knowledge objects and constraints to be edited. In the baseline methods, these steps are also performed by an LLM; our semantic-augmented SGR module merely refines this process. Moreover, due to the sparsity of the constructed knowledge graph, the additional computational overhead remains limited.
>
> > ***(Weakness2)***  *The entire framework's performance hinges on the ability of a foundational LLM to accurately extract entities and relations to form the initial knowledge graph. The paper does not sufficiently discuss how errors or biases from this initial step might propagate through the system and affect the final editing performance.*
>
> > ***(Question 2)***  *The quality of the initial knowledge graph is fundamental to CAKE's success. How does the framework handle inaccuracies, such as missed entities or incorrect relations, from the LLM-based graph extraction step? Have you studied how the quality of this initial graph correlates with final editing performance?*
>
> Thank you for your valuable suggestions. Through manual evaluation, we verified that CAKE's initial knowledge graph accurately and comprehensively covers nearly all knowledge entities present in the original unstructured passage. We report the corresponding evaluation results in the table below. We conducted a human evaluation from two perspectives: extraction accuracy and knowledge completeness.
>
> ||||
> |-|-|-|
> ||knowledge completeness| extraction accuracy|
> |CounterFact|99%|95%|
> |MQ-CF|99%|97%|
> |WikiUpdate|93%|95%|
>
> As can be seen, CAKE's IKG achieves near-perfect coverage of the entities and relations present in the raw text. This strong performance mainly stems from (i) CAKE’s carefully designed, task-specific prompting rules and (ii) the capabilities of the employed LLM (for models with limited intrinsic extraction capacity, such as GPT2-XL, we leverage the GPT-4 API for IKG construction).
>
> > ***(Weakness3)***  *The conflict detection formula includes several hyperparameters (α, β, γ) that are crucial for determining whether two pieces of knowledge are in conflict. The authors provide default values but do not present an analysis of how sensitive the model's performance is to variations in these values, which is important for understanding the framework's robustness.*
>
> > ***(Question 3)***  *The conflict detection mechanism relies on hyperparameters α, β, and γ. How did you arrive at the default value of 0.8 for each? How sensitive is the framework's performance, particularly the trade-off between reliability and locality, to changes in these values?*
>
> Thank you for your valuable suggestions. As the values of  ($\alpha, \beta, \gamma$)  vary, CAKE exhibits reasonable performance fluctuations within a certain range, demonstrating overall robustness. We have provide an ablation study on the impact of the hyperparameters $\alpha, \beta, \gamma$ in Appendix D.1 "*Ablation on Conflict-aware Hyperparameters*".
>
> Specifically, we dynamically vary the ratios $\alpha/\gamma$ and $\beta/\gamma$ in Equation (7) and report the corresponding editing accuracy in Figure 4. The results show that increasing $\alpha/\gamma$ and $\beta/\gamma$ lowers the threshold for identifying knowledge conflicts, leading to more knowledge being classified as conflicting and thus isolated into different experts. This enhances editing reliability, albeit with a slight increase in the number of LoRA modules and associated parameters.
>
> > ***(Weakness 4)***  *The mechanism for choosing between the original model and the edited parameters during inference is described as a "memory routing mechanism." However, the paper lacks specific details about how this router is implemented, how activation memories are stored, or how the similarity threshold is determined, making this critical component difficult to reproduce or evaluate.*
>
> > ***(Question 4)*** *Could you please elaborate on the implementation of the "memory routing mechanism" used at inference time? Specifically, how are the "activation memories" for edited knowledge created and stored, and what method is used to set the similarity threshold for engaging the LoRA experts? *
>
> Thank you for the reviewer's question. In our method, the "activation-similarity routing" is not a core design component, as the underlying idea is relatively common; therefore, we do not emphasize it in our presentation. The "activation-similarity routing" adopt a hard activation-based routing strategy to determine whether an input falls within the scope of edited knowledge.
>
> Specifically, during editing, we retain low-bit lightweight activation vectors for each edited knowledge. At inference time, we compare the cosine similarity between the input prompt and these stored activations. If the similarity exceeds a predefined threshold (Wich is set to 0.7), the input is considered relevant to the edited knowledge, and the corresponding LoRA module is activated for inference. Otherwise, the model relies solely on its original parameters. We will append the Algorithm of this routing in the final version.
>
> The table below presents the impact of the threshold on editing performance on the CounterFact dataset on llama3.2. As the threshold decreases, editing reliability decreases while locality improves. This is because a lower threshold imposes a stricter criterion for selecting LoRA experts, causing the model to rely more heavily on its original parameters during inference. We will report these experimental results on the effect of the threshold in the final version of the paper.
>
> | threshold   | 0.4   | 0.5   | 0.6   | 0.7   | 0.8   |
> | ----------- | ----- | ----- | ----- | ----- | ----- |
> | reliability | 14.74 | 24.56 | 41.16 | 43.24 | 45.72 |
> | locality    | 87.09 | 85.51 | 84.61 | 83.67 | 74.13 |
>
> > ***(Question 5)*** *What is the typical number of "colors" (i.e., LoRA experts) required for the datasets in your experiments? How does the system's performance ... ?*
>
> Thank you for your valuable suggestions. The table below summarizes the number of LoRA modules required per instance across different datasets.
>
> |||||
> |-|-|-|-|
> ||CounterFact|MQuAKE-CF|WIKIupdate|
> |average LoRA nums|5.5|4.8|9.5|
>
> Besides, as the textual input becomes more complex and the number of knowledge conflicts increases, the editing difficulty rises, leading to a slight decline in system performance and a marginal increase in the model's additional parameter count.
>
> In Section 3.4 of the paper, we analyse how editing performance varies with both input-text complexity and the number of knowledge conflicts, as illustrated in Figure 3. The table below further reports how the system's parameter count changes under these same two factors.
> We observe that, as text complexity rises, performance declines slightly, yet our method still maintains a substantial improvement over the baseline. The parameter count increases only marginally—still less than 0.1 % of the original model—because the passage contains more facts, raising the difficulty of editing. Similarly, when more knowledge conflicts are present, more LoRA experts are allocated, leading to a small parameter increase and a modest drop in performance. Notably, text complexity and conflict count are often correlated: more complex passages typically harbour more conflicts, jointly amplifying the editing challenge.
>
> > ***(Question 6)*** *Your ablation study shows that using a single expert for all edits is suboptimal. To further isolate the benefit of your conflict-aware strategy, have you considered comparing the graph-coloring allocation to a simpler baseline, such as randomly assigning conflicting edits to a set of available experts?*
>
> Thank you for your valuable suggestions. The performance of  conflict-aware strategy is significantly better than that of random assignment. We compare our CKE module’s expert-assignment policy with a baseline that randomly assigns conflicting edits to any available expert. We conducted experiments on the CounterFact dataset on llama3.2-3b. The results are shown below：
>
> ||||||
> |-|-|-|-|-|
> ||without CKE|Router Expert|Random Expert|CAKE|
> |editing reliability|34.7|26.04|39.92|43.24|
>
> As can be seen, our method substantially outperforms the random assignment strategy, confirming that the proposed graph-coloring allocation effectively isolates editing conflicts and improves overall editing performance.
>
> We have carefully addressed the comments point by point, and we hope these revisions will further meet your expectations and gain your approval.

---

> > ### Comment · Reviewer_R5dq · 2025-08-05
> >
> > Thank you for the response.
> > I still have concerns regarding evaluation. While the 'Reliability' metric tests for correctness, it doesn't directly measure the 'Generality' of the edited knowledge, which is a metric commonly used in knowledge editing research.  And the paper does not specify how to handle possible new edits. If a new edit request comes, how are inter-edit conflicts (i.e., the new fact conflicting with a previously edited one) detected and resolved? It is unclear whether the graph is dynamically managed when new edits are needed, or if the previously edited samples should all be stored and used with the new edits to construct a new graph. This also directly affects how the LoRA experts should be managed. It is essential for evaluating the practical scalability of the proposed method.

---

> > > ### Author Response · Authors · 2025-08-05
> > >
> > > Thank you very much for your valuable suggestions.
> > >
> > > **About Evaluation.**
> > > We evaluate our method using the standard benchmark designed for wild unstructured text knowledge editing (WUKE) [1].           Unlike traditional model editing benchmarks where the test queries are identical of the original edit inputs in reliability, WUKE evaluates edits reliability using abstract questions derived from unstructured textual paragraphs.           These queries are different from the original edit inputs in expression and serve as a semantic generalization over the full context of the edited knowledge.           As a result, editing reliability in WUKE simultaneously reflects both edit success and generalization.    In contrast, conventional model editing separately measures reliability using the exact original queries, and generalization using their paraphrases.   In the unstructured setting, however, the evaluation query naturally captures both aspects, thereby enabling a more comprehensive assessment of a model’s ability to internalize and generalize knowledge edits.
> > >
> > > **About New Edits.**
> > > This work presents the first approach to knowledge editing on wild unstructured text.           As such, extending to multiple sequential edits is not our primary focus.           Nevertheless, our method is inherently extensible.           Specifically, we can compare new samples to the activation vectors of previously edited samples stored via the memory routing mechanism, allowing us to detect potential edit conflicts without maintaining an explicit knowledge graph.           By quantifying the conflict level of each new edit, we can allocate it to a conflict-free existing LoRA subspace or instantiate a new one.           Given that unstructured text paragraphs often cover distinct topics, semantic overlap—and thus edit conflict—is generally limited.           Therefore, many new edits can effectively reuse prior LoRA modules.           This property enables our method to scale to multiple edits with minimal additional parameters.           We leave a deeper investigation of sequential editing in unstructured settings to future work.
> > >
> > > We sincerely appreciate your valuable feedback, and we hope these revisions further align with your expectations and earn your approval.
> > >
> > > [1] Akew: Assessing knowledge editing in the wild. EMNLP 2024

---

> > > > ### Comment · Reviewer_R5dq · 2025-08-06
> > > >
> > > > Thanks for your response. I will raise the score to 4.

---

> > > > > ### Author Response · Authors · 2025-08-06
> > > > >
> > > > > Thank you for your valuable feedback! We appreciate your constructive reviews for improving our work.

---

### Official Review · Reviewer_7oCG · 2025-07-02

**Clarity:** 2
**Significance:** 3
**Originality:** 3
**Rating:** 5
**Confidence:** 4

**Summary:**

The paper addresses the challenge of “wild, unstructured knowledge editing” (WUKE) for large language models—updating facts when supervision comes only from raw text—by proposing CAKE, a two-stage framework that first derives a Semantic-augmented Graph Representation (SGR) of each paragraph through LLM-prompted extraction, multi-hop inference, and paraphrase diversification, then performs Conflict-aware Knowledge Editing (CAKE) by treating each triple as a node in a conflict graph, colouring that graph with the Welsh–Powell algorithm, and mapping each colour to an orthogonal LoRA sub-space so conflicting edits do not interfere during fine-tuning. Tested on CounterFact, WikiUpdate, and MQuAKE-CF with GPT-2-XL and Llama-3 (3 B & 8 B), CAKE improves reliability by 11–21 percentage points over strong baselines while preserving locality, establishing the first conflict-aware framework for unstructured text edits, introducing a richer graph encoding of paragraph knowledge, and offering an elegant, parameter-efficient gradient-isolation mechanism via multi-expert LoRA.

**Questions:**

1. Equation 7 uses fixed thresholds (α, β, γ) and a single hidden layer to compute conflict edges; how robust are results to these choices?
2. The paper briefly mentions “activation-similarity routing”; what exact algorithm, similarity metric, and thresholds are used, and how does averaging vs. hard selection affect results?

**Ethical Concerns:**

["NO or VERY MINOR ethics concerns only"]

**Final Justification:**

The authors addressed most of my concerns. I raised the score.

**Limitations:**

Yes

**Paper Formatting Concerns:**

No issues.

**Quality:**

3

**Strengths And Weaknesses:**

Strengths:
1. Clear motivation
Editing directly from unstructured text is both realistic and under-explored; the paper articulates why traditional triple-based editors fail and illustrates the two core difficulties with concrete examples (Fig. 1).
2. Methodological originality
SGR cleverly blends LLM-prompted extraction, reasoning, and paraphrasing to densify the knowledge graph, a departure from pipeline OpenIE. CKE uses a classical colouring algorithm to allocate orthogonal LoRA experts—an elegant, lightweight way to resolve conflicting updates.
3.  Solid empirical gains
Reliability improvements (up to +20.9 pp on Llama-3 3 B) are large, consistent across datasets, and ablations isolate the contribution of each module (Table 2).

Weaknesses
1. Scalability concerns
Welsh-Powell is O(|E| log |E|); constructing and colouring a graph for thousands of triples per edit may bottleneck. Suggestion: discuss complexity, memory footprint, and potential batching or heuristic colourings for very large documents.
2. Clarity & writing
Some sections (e.g., 2.2 “directed multigraph G′”) use dense prose and could benefit from concrete pseudo-code. Typographical errors (“editing confilicts” l. 86) and inconsistent notation (rij vs. r′ij) should be fixed.

---

> ### Author Rebuttal · Authors · 2025-07-31
>
> ## Response to Reviewer 7oCG
>
> We appreciate the reviewer for the detailed comments. We have carefully followed up on and settled every issue raised by them.
>
> > ***(Weaknesses 1)*** *Scalability concerns Welsh-Powell is O(|E| log |E|); constructing and colouring a graph for thousands of triples per edit may bottleneck. Suggestion: discuss complexity, memory footprint, and potential batching or heuristic colourings for very large documents.*
>
> Thank you for your valuable suggestions. The memory overhead and computational complexity introduced by our method during the graph construction and coloring process are minimal. We present the complexity and memory footprint of our full method compared to the baseline method  (which removes the conflict graph construction and Welsh-Powell coloring process) on the CounterFact dataset on llama3.2:
>
> | Method             | Average memory (MB) | Editing time (s) |
> | ------------------ | ------------------- | ---------------- |
> | baseline           | 18319.80            | 50.81            |
> | ours               | 18319.83 (+0.0001%) | 55.27 (+8.7%)    |
> | baseline-long text | 18319.90            | 405.76           |
> | ours-long text     | 18319.91 (+0.0001%) | 422.91 (+4.2%)   |
>
> As shown, the graph construction and coloring steps incur only a 0.0001% increase in memory usage and a negligible rise in computation time. This is primarily due to two factors: (1) the constructed graph is sparse—even in large documents containing thousands of triplets, only a subset of knowledge nodes exhibit conflicts, resulting in limited graph coloring complexity; (2) the conflict graph construction and coloring processes typically account for less than 1% of the total training time per editing iteration, rendering the additional overhead negligible.
>
> We will report these experimental results in the final version of the paper and extend this work to editing tasks over larger documents in future research through strategies such as pre-partitioning of knowledge node groups.
>
> > ***(Weaknesses 2)***  *Clarity & writing Some sections (e.g., 2.2 “directed multigraph G") use dense prose and could benefit from concrete pseudo-code. Typographical errors (“editing confilicts” l. 86) and inconsistent notation (rij vs. r′ij) should be fixed.*
>
> Thank you for your thoughtful and valuable suggestions. We have revised the corresponding sections accordingly and will further review and refine the entire manuscript to ensure clarity and readability.
>
> > ***(Question 1)*** *Equation 7 uses fixed thresholds (α, β, γ) and a single hidden layer to compute conflict edges; how robust are results to these choices?*
>
> Thank you for your valuable suggestions. As the values of  ($\alpha, \beta, \gamma$)  vary, CAKE exhibits reasonable performance fluctuations within a certain range, demonstrating overall robustness. We have provide an ablation study on the impact of the hyperparameters $\alpha, \beta, \gamma$ in Appendix D.1 "*Ablation on Conflict-aware Hyperparameters*".
>
> Specifically, we dynamically vary the ratios $\alpha/\gamma$ and $\beta/\gamma$ in Equation (7) and report the corresponding editing accuracy in Figure 4. The results show that increasing $\alpha/\gamma$ and $\beta/\gamma$ lowers the threshold for identifying knowledge conflicts, leading to more knowledge being classified as conflicting and thus isolated into different experts. This enhances editing reliability, albeit with a slight increase in the number of LoRA modules and associated parameters.
>
> Additionally, regarding the impact of the number of edited hidden layers on performance, we follow prior studies[1,2] which observe that a single FFN layer is typically sufficient to encode newly introduced knowledge by selecting a single FFN layer in the middle-to-late stages of the model for editing.  Further increasing the number of edited layers often leads to a degradation in the model's original capabilities and a reduction in editing locality. We will include the corresponding experimental results in the final version of the paper.
>
> > ***(Question 2)*** *The paper briefly mentions “activation-similarity routing”; what exact algorithm, similarity metric, and thresholds are used, and how does averaging vs. hard selection affect results?*
>
> Thank you for the reviewer's question. In our method, the "activation-similarity routing" is not a core design component, as the underlying idea is relatively common; therefore, we do not emphasize it in our presentation. The "activation-similarity routing" adopt a hard activation-based routing strategy to determine whether an input falls within the scope of edited knowledge.
>
> Specifically, during editing, we retain low-bit lightweight activation vectors for each edited knowledge. At inference time, we compare the cosine similarity between the input prompt and these stored activations. If the similarity exceeds a predefined threshold (Wich is set to 0.7), the input is considered relevant to the edited knowledge, and the corresponding LoRA module is activated for inference. Otherwise, the model relies solely on its original parameters. We will append the Algorithm of this routing in the final version.
>
> The table below presents the impact of the threshold on editing performance on the CounterFact dataset on llama3.2. As the threshold decreases, editing reliability decreases while locality improves. This is because a lower threshold imposes a stricter criterion for selecting LoRA experts, causing the model to rely more heavily on its original parameters during inference. We will report these experimental results on the effect of the threshold in the final version of the paper.
>
> | threshold   | 0.4   | 0.5   | 0.6   | 0.7   | 0.8   |
> | ----------- | ----- | ----- | ----- | ----- | ----- |
> | reliability | 14.74 | 24.56 | 41.16 | 43.24 | 45.72 |
> | locality    | 87.09 | 85.51 | 84.61 | 83.67 | 74.13 |
>
> [1] Locating and editing factual associations in gpt. NeurIPS 2022
>
> [2] Wise: Rethinking the knowledge memory for lifelong model editing of large language models. NeurIPS 2024

---

### Comment · Area_Chair_w3Vo · 2025-08-04
**Friendly Reminder to Acknowledge or Update Your Review**

Dear Reviewers,

Thank you for your time and effort in reviewing the submissions and for providing valuable feedback to the authors.

If you haven’t already done so, we kindly remind you to review the authors’ rebuttals and respond accordingly. In particular, if your evaluation of the paper has changed, please update your review and explain the revision. If not, we would appreciate it if you could acknowledge the rebuttal by clicking the “Rebuttal Acknowledgement” button at your earliest convenience.

This step ensures smooth communication and helps us move forward efficiently with the review process.

We sincerely appreciate your dedication and collaboration.

Best regards, AC

---

### Note · Authors · 2025-08-14

Dear Area Chairs, Program Chairs, and Reviewers,

We sincerely appreciate your dedication in evaluating our paper and providing constructive feedback. As recognized in all reviewers' comments, our paper is well-motivated and innovative, effectively tackling the challenge of editing LLMs using real-world, unstructured text, while demonstrating significant improvements across a wide range of benchmarks.

During the discussion phase, **we have comprehensively addressed all concerns raised by the reviewers and obtained their approval.** The key points are as follows:

1.Computational complexity and memory usage of the proposed method:

Through experimental results and analyses, we have demonstrated that the increase in computational complexity and memory usage brought by our proposed method compared to the baseline model is minimal and almost negligible. (Response to Reviewer 7oCG, R5dq)

2.Analysis of the impact of hyperparameters on the method:

We have provided an ablation study on the impact of hyperparameters in Appendix D.1 "Ablation on Conflict-aware Hyperparameters". Experiments show that the proposed method exhibits reasonable performance fluctuations within a certain range, demonstrating overall robustness. (Response to Reviewer R5dq, 7oCG, NqoV)

3.The implementation of the "memory routing mechanism":

We have elaborated on this mechanism in the discussions with reviewers and gained their recognition. (Response to Reviewer R5dq)

We remain fully committed to incorporating all constructive feedback into our final manuscript. We fully recognize the demanding nature of the review process and sincerely appreciate the considerable time and effort invested by the Area Chairs and Program Chairs in overseeing this process. Thank you for your continued support and for your fair, objective, and professional oversight.

---

### Decision · Program_Chairs · 2025-09-17

**Decision:**

Accept (spotlight)

**Comment:**

This paper presents a novel framework for conflict-aware knowledge editing (CAKE) of LLMs from unstructured text. CAKE addresses two key challenges: representation ambiguity, resolved via a semantic-augmented graph representation, and editing conflicts, handled through graph coloring to isolate conflicting edits into orthogonal LoRA subspaces. Extensive experiments on 3 datasets with different models show consistent improvements in reliability, demonstrating that CAKE effectively updates LLM knowledge without compromising unrelated facts.

**Strengths (and Reasons to Accept):**
- The paper addresses the practical challenge of editing LLMs using raw text rather than structured triples, which is under-explored and highly relevant.
- Semantic-augmented graph representation densifies knowledge representation via LLM-prompted extraction, reasoning, and paraphrasing. The graph-coloring-based allocation of LoRA experts also resolves conflicts elegantly.
- Improvements are consistent across multiple datasets and models. Ablation studies confirm the contributions of SGR and CKE, and comparisons against strong baselines highlight the effectiveness of the approach.
- The workflow, algorithms, and experimental setup are generally well explained.

**Weaknesses:**
- Scalability: Graph construction and coloring could be computationally expensive for very large documents. Complexity analysis or discussion of potential heuristics would strengthen the paper.
- Locality performance: CAKE sometimes underperforms baselines on the locality (Loc.) metric; additional discussion would clarify underlying causes.

The authors spent tremendous efforts during the rebuttal period addressing reviewers' concerns. Two reviewers raised the score. Finally, all reviewers are unanimously inclined to the positive side. The methodological novelty, solid empirical results, and clear presentation outweigh the minor concerns. Therefore, I recommend **accept**.